# Effect of intermittent preventive treatment for malaria with dihydroartemisinin-piperaquine on immune responses to vaccines among rural Ugandan adolescents: randomised controlled trial protocol B for the 'POPulation differences in VACcine responses' (POPVAC) programme

AN, GN and LZ contributed equally.

► http://dx.doi.org/10.1136/bmjopen-2020-040425
► http://dx.doi.org/10.1136/bmjopen-2020-040426

For numbered affiliations see end of article.

**Correspondence to**
Dr Gyaviira Nkurunungi; Gyaviira.Nkurunungi@mrcuganda.org

Agnes Natukunda,[1] Gyaviira Nkurunungi [ORCID],[1] Ludoviko Zirimenya,[1] Jacent Nassuuna,[1] Gloria Oduru,[1] Rebecca Amongin,[1] Prossy N Kabuubi,[1] Alex Mutebe,[1] Caroline Onen,[1] Susan Amongi,[1] Esther Nakazibwe,[1] Florence Akello,[1] Samuel Kiwanuka,[1] Fred Kiwudhu,[1] Moses Sewankambo,[1] Denis Nsubuga,[1] Robert Kizindo,[1] Sarah G Staedke,[2,3] Stephen Cose,[1,2] Emily Webb,[4] Alison M Elliott,[1,2] on behalf of the POPVAC trial team

## ABSTRACT

**Introduction** Drivers of lower vaccine efficacy and impaired vaccine-specific immune responses in low-income versus high-income countries, and in rural compared with urban settings, are not fully elucidated. Repeated exposure to and immunomodulation by parasite infections may be important. We focus on *Plasmodium falciparum* malaria, aiming to determine whether there are reversible effects of malaria infection on vaccine responses.

**Methods and analysis** We have designed a randomised, double-blind, placebo-controlled, parallel group trial of intermittent preventive malaria treatment versus placebo, to determine effects on vaccine response outcomes among school-going adolescents (9 to 17 years) from malaria-endemic rural areas of Jinja district (Uganda). Vaccines to be studied comprise BCG vaccine on day 'zero'; yellow fever, oral typhoid and human papilloma virus vaccines at week 4; and tetanus/diphtheria booster vaccine at week 28. Participants in the intermittent preventive malaria treatment arm will receive dihydroartemisinin/piperaquine (DP) dosed by weight, 1 month apart, prior to the first immunisation, followed by monthly treatment thereafter. We expect to enrol 640 adolescents. Primary outcomes are BCG-specific interferon-γ ELISpot responses 8 weeks after BCG immunisation and for other vaccines, antibody responses to key vaccine antigens at 4 weeks after immunisation. In secondary analyses, we will determine effects of monthly DP treatment (versus placebo) on correlates of protective immunity, on vaccine response waning, on whether there are differential effects on priming versus boosting immunisations, and on malaria infection prevalence. We will also conduct exploratory immunology assays among subsets of participants to further characterise effects of the intervention on vaccine responses.

## Strengths and limitations of this study

► This will be the first well-powered trial to evaluate the effect of repeated intermittent presumptive anti-malarial treatment on vaccine response outcomes in adolescents.

► Effects on both live-attenuated and inert vaccines will be studied.

► The use of an interventional research design will contribute strong evidence to determine whether malaria has causal and reversible effects on vaccine response outcomes, and our strong immunoepidemiological design and nested immunological studies will make it possible to address specific hypotheses regarding pathways of effects.

► The use of vaccines most of which are part of Uganda's national Expanded Programme on Immunisation, makes the findings from this study applicable and could inform the introduction of combined school health programmes for malaria control and vaccine provision.

► With the high prevalence of malaria in the proposed study area, participants in the placebo arm may notice more frequent episodes of malaria and this could lead to individual self-unblinding and self-treatment, compromising the design of the study.

**Ethics and dissemination** Ethics approval has been obtained from relevant Ugandan and UK ethics committees. Results will be shared with Uganda Ministry of Health, relevant district councils, community leaders and study participants. Further dissemination will be done through conference proceedings and publications.

**Trial registration number** Current Controlled Trials identifier: ISRCTN62041885.

## INTRODUCTION

Population differences in vaccine-specific immune responses and efficacy have been documented. In tropical low-income, rural settings, responses to both live[1–6] and non-live[7 8] vaccines are often impaired, compared with urban and high-income country settings. Studies have also shown that responses to candidate vaccines (such as those to tuberculosis,[9] malaria[10] and Ebola[11]) are lower in Africa than in Europe or America. Drivers of these population differences in vaccine response are incompletely understood. While genetic differences may play a role, they cannot fully explain this phenomenon: for example, migrant and native English populations have comparable BCG vaccine efficacy.[12] Previous exposure to the pathogen targeted by the vaccine, or to related organisms, may mask the benefit of the vaccine;[13 14] however, this mechanism cannot explain observations for vaccines against rare organisms such as Ebola.[11] The role of immunomodulating environmental factors has been discussed,[2] but evidence is inconclusive. Parasites, particularly, have long been proposed as modulators of vaccine responses,[15 16] but this has not been fully substantiated in well-powered trials aimed at evaluating reversibility of their effects in human populations.[17]

This study is one of three parallel trials whose designs and cross-cutting analyses are described separately in this journal (bmjopen-2020-040425, bmjopen-2020-040426 and bmjopen-2020-040430). The focus of this POPulation differences in VACcine responses (POPVAC) B trial is on infection with malaria parasites. Malaria remains highly prevalent in Africa and Uganda is among the six countries with the highest prevalence of malaria parasites on the continent.[18] A recent nationally representative survey estimated parasite prevalence (based on microscopy) of 32% in children aged 8 to 10 years, with wide variability by region.[19] Emerging insecticide and drug resistance threatens gains towards control.[20] Interventions like intermittent preventive treatment (IPT) are recommended by the WHO for use in preventing malaria in pregnant women, infants and young children living in areas of seasonal malaria transmission.[20] In a recently concluded trial in Uganda, IPT in schoolchildren was shown to be beneficial in protecting individual children and reducing community transmission of malaria.[21 22]

Several studies have previously reported impaired meningococcal, tetanus and typhoid vaccine responses in children infected with malaria parasites.[23–26] Another study showed that children receiving malaria chemoprophylaxis had better responses to meningococcal vaccine than children not receiving malaria chemoprophylaxis.[27]

In a Ugandan study, malaria infection in mothers during pregnancy, and in their infants, was associated with reduced antibody responses to measles immunisation in infancy.[28] We have previously observed that infant malaria infection is associated with reduction in BCG and tetanus vaccine-specific responses.[29] On the other hand, malaria infection could improve responses to some vaccines and a study by Brown *et al*[30] reported increased responses to a human papilloma virus (HPV) vaccine among individuals with malaria parasitaemia. In a 2010 review, Cunnington and Riley showed that, despite long-standing interest, the impact of malaria on vaccine responses was not adequately understood.[16] Although there is little evidence that malaria impairs responses to protein vaccines given in multiple doses,[16 30 31] they found consistent evidence of an adverse effect on the response to polysaccharide vaccines.[16]

Malaria may impact vaccine responses due to acute immunological changes (associated with fever),[32] or by longer term effects (for example on T follicular helper cell and B cell function).[33] The extent to which infection with malaria impacts immunological characteristics associated with vaccine responses may best be determined by intervention studies. This trial protocol B of the 'POPVAC' programme has been designed to evaluate the effect of IPT of malaria on vaccine responses. We summarise the protocol here. This study is one of three parallel trials whose designs and cross-cutting analyses are described separately in this issue. Understanding the predictors of vaccine responses, and the factors that drive them, will contribute to finding ways of improving vaccine efficacy for rural, tropical settings.

## HYPOTHESIS

The overarching goal of the POPVAC programme is to understand population differences in vaccine responses in Uganda, in order to identify strategies through which vaccine effectiveness can be optimised for the low-income, tropical settings where they are especially needed. For this Trial B, we focus on the hypothesis that malaria infection suppresses responses to unrelated vaccines, and that this effect can be reversed, at least in part, by IPT of malaria in schools in high transmission settings.

## OBJECTIVE

To determine whether there are reversible effects of malaria infection on vaccine response in adolescents, using an intervention study.

## METHODS AND ANALYSIS
### Design, setting and participants

Standard Protocol Items: Recommendations for Interventional Trials (SPIRIT) reporting guidelines[34] are used. We will conduct an individually randomised, double-blind, placebo-controlled, parallel group trial of

monthly dihydroartemisinin/piperaquine (DP) versus placebo among primary school children in rural schools in Jinja district, Uganda. We will recruit 640 participants, randomising 320 to each trial arm. The study cohort will recruit participants aged 9 to 17 years from primary school year 1 up to 6 (to avoid primary leaving examinations in late year 7, and loss to follow-up of children who leave from primary 7). Schools will be selected purposefully to assure malaria prevalence appropriate to our design. The study setting is remote from Lake Victoria and the River Nile, in order to minimise exposure to schistosomiasis.

## Recruitment criteria
### Inclusion criteria
i.   Attending the selected school and planning to continue to attend the school for the duration of the study.
ii.  Aged 9 to 17 years and enrolled in primary 1 to 6.
iii. Written informed assent by participant and consent by parent or guardian.
iv.  Agree to avoid pregnancy for the duration of the trial (female only).
v.   Willing to provide locator information and to be contacted during the course of the trial.
vi.  Able and willing (in the investigator's opinion) to comply with all the study requirements.

### Exclusion criteria
i.   Clinically significant history of immunodeficiency (including HIV), cancer, cardiovascular disease, gastrointestinal disease, liver disease, renal disease, endocrine disorder or neurological illness.
ii.  Moderate or severe acute illness characterised by any of the following symptoms: fever, impaired consciousness, convulsions, difficulty in breathing or vomiting; or as determined by the attending project clinician.
iii. Family history of sudden death attributable to heart condition in a first-degree relative.
iv.  Family history of long QT syndrome.
v.   Know congenital prolongation of the corrected QT (QTc) interval.
vi.  History of known heart disease or fainting.
vii. Known allergy or history of adverse reaction to DP or to artemether-lumefantrine.
viii. History of serious psychiatric condition or disorder.
ix.  Previous immunisation with yellow fever (YF), oral typhoid or HPV vaccine; previous immunisation with BCG or tetanus and diphtheria vaccine (Td) at age ≥5 years.
x.   Concurrent oral or systemic steroid medication or the concurrent use of other immunosuppressive agents within 2 months prior to enrolment.
xi.  Current use of medications known to prolong the QT interval.
xii. History of allergic reaction to immunisation or any allergy likely to be exacerbated by any component of the study vaccines including egg or chicken proteins.
xiii. Tendency to develop keloid scars.

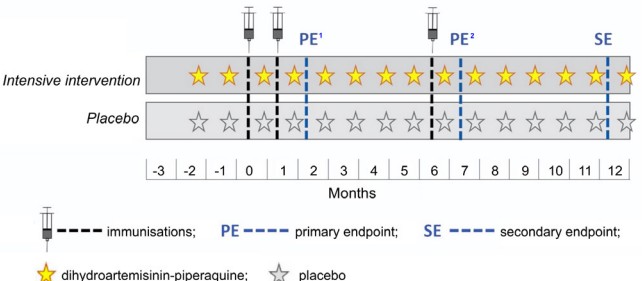

**Figure 1** Outline of immunisations and interventions. [1]PE will be at 8 weeks post BCG and 4 weeks post yellow fever (YF-17D), oral typhoid (Ty21a), human papilloma virus and tetanus/diptheria (Td) vaccination. [2]PE for responses to Td given at 28 weeks.

xiv. Haemoglobin less than 80 g/L.
xv.  Positive HIV serology.
xvi. Positive pregnancy test.
xvii.Female currently lactating, confirmed pregnancy or intention to become pregnant during the trial period.
xviii.   Use of an investigational medicinal product or non-registered drug, live vaccine, or medical device other than the study vaccines for 30 days prior to dosing with the study vaccine, or planned use during the study period.
xix. Administration of immunoglobulins and/or any blood products within the 3 months preceding the planned trial immunisation date.

Further information on recruitment criteria can be found in online supplemental information.

## Interventions
We will individually randomise participants in a 1:1 ratio to an experimental intervention of monthly DP versus a DP-matching placebo in a double-blind, placebo-controlled trial (figure 1 and online spplemental table S1). DP has been shown to reduce prevalence of asymptomatic malaria by 94% and malaria incidence by 96% among Ugandan school children.[35] The DP treatment arm will receive two doses, 1 month apart, prior to the first immunisation, followed by monthly treatment thereafter. Each dose will be calculated based on the participant's weight (in kg).[36] DP (dihydroartemisinin 40 mg+piperaquine phosphate 320 mg) tablets will be administered once a day for 3 consecutive days. The standard arm will receive placebo since, as yet, routine preventive malaria treatment in schools is not a Uganda Ministry of Health policy.

## Randomisation and allocation to treatment arm
An independent statistician will generate the randomisation code using a randomly permuted block size. This code will be embedded into a web-based randomisation system in REDCap (Research Electronic Data Capture) software.[37 38] At enrolment, eligibility criteria will be checked and eligible participants will be allocated sequentially to the next randomisation number, with the corresponding trial arm designated in REDCap. The

**Table 1** Immunisation schedule

| | Immunisation week 0 | Immunisation week 4 | Immunisation week 8 | Immunisation week 28 | Immunisation week 52 |
|---|---|---|---|---|---|
| Live vaccines | BCG vaccination/re-vaccination* | Yellow fever (YF-17D) Oral typhoid (Ty21a) | | | |
| Non-live vaccines | | HPV prime† | HPV boost for girls aged ≥14 years‡§ | HPV boost† and Td boost | Td boost§¶ |

*Prior BCG status may vary (data on history and documentation of prior BCG, and presence of a BCG scar, will be documented although these approaches have limitations for determining BCG status)
†Both girls and boys will receive the HPV vaccine
‡The National EPI programme recommends three doses of HPV vaccine for older girls
§These doses will be given to comply with guidelines but outcomes specifically relating to these doses will not be assessed.
¶Priming by immunisation in infancy is assumed.
EPI, Expanded Programme on Immunisation; HPV, human papilloma virus; Td, Tetanus/diphtheria.

randomisation code will be kept securely by the trial statistician with a second copy held by a data manager or statistician not otherwise involved in the trial at the Medical Research Council (MRC)/Uganda Virus Research Institute (UVRI) and London School of Hygiene and Tropical Medicine (LSHTM) Uganda Research Unit.

### Blinding
The trial will be double-blind, and will remain so until data collection and data cleaning are complete and data locked for analysis, unless unblinding is required for individual participants who become pregnant (for these cases unblinding can be requested by the Principal Investigator) or as a result of serious adverse events or reactions (at the request of the Data and Safety Monitoring Board).

### Immunisations
We will study a portfolio of licensed vaccines (live and inert, oral and parental, priming and boosting) expected to be beneficial (in some cases, already given) to adolescents in Uganda. Our schedule (table 1 and online supplemental table S1) will comprise three main immunisation days (week 0, week 4 and week 28). Additional HPV immunisation will be provided for girls aged 14 years or above, and a second Td boost will be given after completion of the study, to accord with the national Expanded Programme on Immunisation (EPI) routines but the response to these will not specifically be addressed. Further rationale for the selection of vaccines is detailed in online supplemental information. Our schedule has been developed in consultation with the EPI programme and is cognisant of potential interference between vaccines (online supplemental information).

### Schedule of immunisation and sampling
The schedule of immunisation and sampling is outlined in figure 1 and online supplemental table S1. While optimal timings for outcome measures vary between vaccines, sampling at 8 weeks post BCG and 4 weeks post YF-17D, Ty21a, HPV and Td is proposed for the primary endpoints, targeting the establishment of memory responses and approximate peak of antibody responses.[39–43] A secondary endpoint at 1 year will assess waning. All analyses will take baseline measurements into account. Immunisation post-ponement criteria are detailed in online supplemental information.

### Outcomes
#### Primary outcomes
These will be assessed in all participants.
i.   BCG: BCG-specific interferon-γ ELISpot response 8 weeks post BCG immunisation.
ii.  YF-17D: neutralising antibody titres (plaque-reduction neutralisation test) at 4 weeks post YF immunisation.
iii. Ty21a: *Salmonella typhi* lipopolysaccharide-specific IgG concentration at 4 weeks post Ty21a immunisation.
iv.  HPV: IgG specific for L1-proteins of HPV-16/18 at 4 weeks post HPV priming immunisation.
v.   Td: Td-specific IgG concentration at 4 weeks post Td immunisation.

#### Secondary outcomes
These will be assessed in all participants and will further investigate estimates of protective immunity (for vaccines where these are available) and dynamics of the vaccine responses, as well as the impact of the interventions on parasite clearance.

##### Protective immunity
Proportions with protective neutralising antibody (YF); protective IgG levels (TT);[44] seroconversion rates (Ty21a) at 4 weeks post the corresponding immunisation.

##### Response waning
Primary outcome measures (all vaccines) repeated at week 52, and area under the curve analyses.

##### Priming versus boosting
Effects on priming versus boosting will be examined for HPV only, comparing outcomes 4 weeks after the first, and 4 weeks after the second vaccine dose.

### Current malaria infection status and intensity

Will be assessed retrospectively by PCR on stored samples collected on immunisation days and at week 52.

Furthermore, our sample collection will offer opportunities for an array of exploratory immunological evaluations on stored samples, focussing mainly on vaccine antigen-specific outcomes. Exploratory assessments will provide further detail on immune response characteristics over the study time-course, and on the role of immunological profiles in malaria-mediated modulation of vaccine-specific responses.

### Additional evaluation of parasite infection

1. Current *Schistosoma mansoni* infection status and intensity will be determined by serum/plasma levels of circulating anodic antigen (CAA). This method is quantitative, highly specific for *Schistosoma* infection and much more sensitive than the conventional Kato Katz method.[45] CAA will be assessed retrospectively on stored samples collected at baseline and at weeks 28 and 52.
2. Prior exposure to schistosomiasis will be evaluated by ELISA for IgG to schistosome egg antigen using stored blood samples collected at baseline.
3. The presence of other helminth infections will be determined retrospectively using stool PCR of samples collected at baseline and at weeks 28 and 52. In accord with national guidelines, all participants will be treated with albendazole or mebendazole after collection of samples for primary endpoints at week 8 and 28, and after collection of samples for secondary endpoints at week 52.
4. Malarial fever: Individuals presenting with fever will be investigated using rapid diagnostic tests for malaria and treated based on the results and according to prevailing national guidelines.
5. Prior malaria exposure will be evaluated by ELISA for IgG to malaria antigen using stored samples collected at baseline.

### Sample size considerations

Based on the literature[6 46 47] and preliminary data, we anticipate that SD of primary outcome measures will lie between 0.3 and 0.6 $\log_{10}$; that responses in rural, high-parasite settings may be 0.3 to 0.4 $\log_{10}$ smaller than in the urban setting, and that effective treatment of parasitic infections may restore responses by approximately 0.2 $\log_{10}$ (Tweyongyere *et al*).[48] We therefore power our study to detect differences of this magnitude (0.2 $\log_{10}$) or (in some cases) smaller. Among 5 to 15 year olds in this study setting, previous studies have reported malaria prevalence of >50% based on microscopy;[21 22] we assume malaria prevalence of ≥60% based on PCR.

Based on these assumptions, we plan to include 640 participants in total (320 DP and 320 placebo); of whom 384 are expected to be malaria infected, giving 192 participants in each trial arm who are infected at baseline.

Table 2 shows power estimates, for 5% significance level and assuming 20% loss to follow-up.

### ETHICS AND DISSEMINATION

Ethical approval has been granted from the Research Ethics Committees of the UVRI (UVRI REC, reference: GC/127/19/05/681) and the LSHTM (reference: 16033), and from the Uganda National Council for Science and Technology (UNCST, reference: HS 2487) and the Uganda National Drug Authority (NDA, reference: CTC0117/2020). Any protocol amendments will be submitted to ethics committees and regulatory bodies for approval before implementation.

Participants will be adolescents and therefore a vulnerable human population. Care will be taken to provide adequate, age and education-status appropriate information and to ensure that it is understood; and to emphasise that participation is voluntary. Participants will be enrolled only when they have given their own assent and when consent has been given by the parent or guardian. No major risks to the participants are anticipated since all the treatments and vaccines to be given are licensed and known to be safe. The main risk to participants will be time lost from school work: we will work with teachers and parents to minimise disruption to classes, and will avoid enrolment of primary seven students since these classes are involved in national examinations. Further risks are discussed in online supplemental information.

**Table 2** Power estimates (5% significance level)

| Standard deviation ($\log_{10}$) | $\log_{10}$ difference | | | | | | |
|---|---|---|---|---|---|---|---|
| | 0.08 | 0.10 | 0.12 | 0.14 | 0.16 | 0.18 | 0.20 |
| 192 DP vs 192 placebo (malaria infected only) | | | | | | | |
| 0.3 | 65% | 83% | 94% | 98% | >99% | >99% | >99% |
| 0.4 | 42% | 59% | 75% | 87% | 94% | 98% | 99% |
| 0.5 | 29% | 42% | 56% | 69% | 80% | 88% | 94% |
| 0.6 | 21% | 31% | 42% | 53% | 65% | 75% | 83% |

Cells highlighted in grey correspond to >80% power.
DP, dihydroartemisinin/piperaquine.

For this trial, we gave particular consideration to the differential provision of malaria treatment. IPT with DP is an experimental intervention that has been shown to reduce the prevalence of anaemia and reduce episodes of clinical malaria in Ugandan schools[35] and was shown to be safe under suitable exclusion criteria[49] but has not been adopted as standard of care. To manage the expected differential benefits of the interventions for anaemia, a full blood count will be performed at baseline, as discussed above; anaemic children will be managed appropriately and severely anaemic children excluded. Additional safety monitoring during the trial will be done by the Data and Safety Monitoring Board (DSMB). Further information on the DSMB can be found in online supplemental information.

Malaria infection status will be determined retrospectively through assays conducted in bulk on stored samples (malaria PCR). These results will not, therefore, be useful to determine management of individual participants. Participants in the placebo arms will receive lower levels of anti-malaria treatment. However, all trial arms will receive a minimum of well-implemented national standard of care. Malaria standard of care will comprise provision of bed nets to minimise malaria exposure for all participants. Rapid diagnostic tests and treatment will be made readily available for participants who develop symptomatic malaria.

Study findings will be published through open-access peer-reviewed journals, presentations at local, national and international conferences and to the local community through community meetings. Anonymised participant level data sets generated will be available on request.

### Participant and public involvement

Concepts involved in this work have been discussed with colleagues at the Vector Control Division and EPI in the Ministry of Health (Uganda). Before commencing the trial, discussions will be held with officials at Jinja District Council, community leaders and Village Health Teams from sub-counties in which selected schools are located. In addition, we will engage teachers and parents in planning the detailed standard operating procedures for the study. Prior to recruitment, we will hold meetings to explain the proposed work to teachers, parents and students and to address their questions. At the end of the study, results will be shared with these stakeholders and with participants.

### Data management and analysis

Sociodemographic information and clinical and laboratory measurements will be recorded and managed using REDCap (Research Electronic Data Capture) tools,[37 38] with paper-based forms as back-up. All data will be recorded under a unique study ID number. When paper forms must be used, data will be double entered in a study-specific database, with standard checks for discrepancies. All data for analysis will be anonymised and stored on a secure and password-protected server, with access limited to essential research personnel.

The effect of monthly IPT for malaria versus no IPT on the outcomes will be analysed using unpaired t-tests, with results presented as a mean difference in vaccine response measure together with 95% CI and p value. In the event of any imbalance in key characteristics between trial arms, multivariable regression models will be used to adjust for these potential confounders. We anticipate that outcomes will be positively skewed, and will apply log transformations to normalise distributions before analysis is conducted. Information on malaria infection status will only be available after randomisation. The primary analysis will be done on individuals identified as infected at baseline (through the randomisation, these will be balanced between treatment arms); this will test the hypothesis that treating the infection (and subsequent reinfections) removes parasite-induced effects on vaccine responses. Secondary analyses will include all randomised individuals; this will provide insight into the potential benefit of the interventions as public health measures.

## DISCUSSION

Parasites have long been proposed as modulators of vaccine responses;[15 16] malaria is highly prevalent in settings with poor vaccine responses. Drivers of population differences in vaccine response are not fully elucidated; improved understanding is important for effective vaccine development and implementation. If treating malaria improves vaccine responses, combining parasite control with immunisation programmes offers an attractive, practical public health intervention for schools and communities.

This study aims to determine whether treating malaria with DP before and after immunisation with unrelated vaccines can improve vaccine responses in adolescents. The use of DP in this study is considered an attractive option for preventive treatment and preventive chemotherapy for malaria because of the long half-life of piperaquine (approximately 23 days).[49] Despite identified benefits in reducing episodes of malaria and anaemia,[35] DP has not been adopted as standard of care either in Uganda or elsewhere. Although we hypothesise that treatment of malaria will improve vaccine responses, it is possible that it will negatively impact immune responses, or that effects will differ between types of vaccines. This study is expected to add further evidence regarding the potential benefits of monthly DP for school children by determining the effect, whether beneficial or negative, on vaccine responses, as well as collecting data on malaria infection status, thereby further contributing to policy debate in this field.

A concern arises where treatment of malaria with DP may have potential adverse effects such as cardiac toxicity, particularly dose-dependent prolongation of the QTc interval and associated increased risk of arrhythmias. There has been concern that provision of multiple doses

might increase this risk especially when taken with food. However, a recent meta-analysis involving 11 trials of DP (9 IPT and 2 treatment trials) involving 14 628 participants, 3935 of whom received multiple treatments with DP, found no evidence to suggest significant cardiac toxicity.[49] Few of the studies measured electrocardiographic changes, but those that did found no increase in QTc intervals with increasing numbers of doses. Therefore, the use of DP in this study where malaria prevalence is high is not expected to pose a significant risk to the participants. Also, although the use of DP necessitates strict exclusion criteria, based on previous studies in Uganda,[22] it is not anticipated that many participants will meet these exclusion criteria.

We expect about 60% of the adolescents to have malaria infection at the outset in the proposed study area. This may vary by season and year, and the prevalence could be much lower depending on the timing of the start of the study. If this becomes the case, the two study arms will not be different enough to test our hypothesis. To mitigate this, we will work closely with colleagues monitoring trends in malaria prevalence in Uganda and adjust our study site to optimise the likelihood of achieving the expected prevalence.

This interventional study will help us to understand how malaria affects responses to commonly used vaccines, which will inform implementation of public health programmes for immunisation and infection control especially in low-income and middle-income countries.

## Study timeline

Applications for ethical approval were submitted in May 2018, with approval received in September 2018 (UVRI REC), May 2019 (UNCST), June 2019 (LSHTM) and January 2020 (NDA). Collaborator/investigator/trial steering committee meetings were also held during the initial 12-month planning period. Recruitment to POPVAC B is scheduled to commence in February 2021. Intervention will be up to 12 months, with completion of the project scheduled for April 2022.

**Author affiliations**
[1]Immunomodulation and Vaccines Programme, MRC/UVRI and LSHTM Uganda Research Unit, Entebbe, Uganda
[2]Department of Clinical Research, London School of Hygiene and Tropical Medicine, London, UK
[3]Infectious Diseases Research Collaboration, Kampala, Uganda
[4]MRC Tropical Epidemiology Group, Department of Infectious Disease Epidemiology, London School of Hygiene & Tropical Medicine, London, UK

**Acknowledgements** We thank the Uganda National Expanded Programme for Immunisation, Sanofi Pasteur and PaxVax for providing the human papilloma virus, yellow fever and oral typhoid vaccines, respectively. The BCG and tetanus-diphtheria vaccines were kind donations from the Serum Institute of India. We thank the Jinja district local government for their support. We also thank members of the POPVAC programme steering committee (chaired by Professor Richard Hayes) and the Data and Safety Monitoring Board (Dr David Meya, Professor Andrew Prendergast and Dr Elizabeth George).

**Collaborators** POPVAC trial team: principal investigator: Alison Elliott; project leader: Ludoviko Zirimenya; laboratory staff: Gyaviira Nkurunungi, Stephen Cose, Rebecca Amongin, Beatrice Nassanga, Jacent Nassuuna, Irene Nambuya, Prossy Kabuubi, Emmanuel Niwagaba, Gloria Oduru and Grace Kabami; statisticians and data managers: Emily Webb, Agnes Natukunda, Helen Akurut and Alex Mutebe; clinicians: Anne Wajja, Milly Namutebi, Christopher Zziwa and Joel Serubanja; nurses: Caroline Onen, Esther Nakazibwe, Josephine Tumusiime, Caroline Ninsiima, Susan Amongi and Florence Akello; internal monitor: Mirriam Akello; field workers: Robert Kizindo, Moses Sewankambo, Denis Nsubuga, Samuel Kiwanuka and Fred Kiwudhu; boatman: David Abiriga; administrative management: Moses Kizza and Samsi Nansukusa; internal and external collaborators: Pontiano Kaleebu, Hermelijn Smits, Maria Yazdanbakhsh, Govert van Dam, Paul Corstjens, Sarah Staedke, Henry Luzze, James Kaweesa, Edridah Tukahebwa, Elly Tumushabe and Moses Muwanga.

**Contributors** AME conceived the study. AME, AN, GN, ELW, SC, LZ and SGS contributed to study design. LZ, GO, PNK, SA, CO, RA, EN and FA are site clinicians/ nurses/clinical laboratory technicians providing valuable input on clinical considerations of the intervention. SK, FK, DN, MS and RK are field workers handling the organisational integration of the intervention. AN, AM and ELW are involved in organisation of the databases, trial randomisation, treatment allocation and drawing up of analytical plans. AN, GN, LZ, JN, SC, ELW and AME drafted the manuscript. All authors reviewed the manuscript, contributed to it and approved the final version.

**Funding** The POPVAC programme of work is supported by the Medical Research Council of the UK (grant number MR/R02118X/1). SC and JN are supported in part by the Makerere University—Uganda Virus Research Institute Centre of Excellence for Infection and Immunity Research and Training (MUII-plus). MUII-plus is funded under the DELTAS Africa Initiative. The DELTAS Africa Initiative is an independent funding scheme of the African Academy of Sciences (AAS), Alliance for Accelerating Excellence in Science in Africa (AESA) and supported by the New Partnership for Africa's Development Planning and Coordinating Agency (NEPAD Agency) with funding from the Wellcome Trust (grant 107743) and the UK Government. The Medical Reseach Council/Uganda Virus Research Institute and London School of Hygiene and Tropical Medicine Uganda Research Unit is jointly funded by the UK MRC and the UK Department for International Development (DFID) under the MRC/ DFID Concordat agreement and is also part of the EDCTP2 programme supported by the European Union.

**Disclaimer** The study sponsor (London School of Hygiene and Tropical Medicine) and funders had no role in study design; collection, management, analysis and interpretation of data; writing of the protocol; and the decision to submit the protocol for publication.

**Competing interests** AE reports a grant from the Medical Research Council, UK (POPVAC programme funding).

**Patient consent for publication** Not required.

**Provenance and peer review** Not commissioned; externally peer-reviewed.

**ORCID iD**
Gyaviira Nkurunungi http://orcid.org/0000-0003-4062-9105

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
