## [Reviewer comments · BMJ Open]

ARTICLE DETAILS

TITLE (PROVISIONAL)	The effect of intermittent preventive treatment for malaria with dihydroartemisinin-piperaquine on immune responses to vaccines among rural Ugandan adolescents: randomised controlled trial protocol B for the 'POPulation differences in VACcine responses' (POPVAC) programme
AUTHORS	Natukunda, Agnes; Nkurunungi, Gyaviira; Zirimenya, Ludoviko; Nassuuna, Jacent; Oduru, Gloria; Amongin, Rebecca; Kabuubi, Prossy; Mutebe, Alex; Onen, Caroline; Amongi, Susan; Nakazibwe, Esther; Akello, Florence; Kiwanuka, Samuel; Kiwudhu, Fred; Sewankambo, Moses; Nsubuga, Denis; Kizindo, Robert; Staedke, Sarah; Cose, Stephen; Webb, Emily; Elliott, Alison

VERSION 1 – REVIEW

REVIEWER	Odilon Paterne NOUATIN Centre de Recherches Médicales de Lambaréné / GABON Faculté des sciences et techniques /Université d'Abomey Calavi, BENIN
REVIEW RETURNED	12-Jul-2020

GENERAL COMMENTS	1- In the introduction, the choice of malaria as an infection that could affect vaccine responses is not well justified. 2- What is the prevalence of malaria in Uganda in general, and in the study area in particular? It could be mentioned in the introduction 3- Line 82: put "WHO" 4- Line 319: It mentioned: "DP has not been adopted as standard of care either in Uganda or elsewhere." Why then choose this drug for treatment? If for some participants, you observe a failure in the treatment of malaria with this drug, would this not affect your expected results? 5- Line 310: Discussion. "Parasites have long been proposed as modulators of vaccine responses". Please add reference
---

REVIEWER	Jorge Gomez Marin Universidad del Quindío Colombia
REVIEW RETURNED	25-Jul-2020

GENERAL COMMENTS	This is randomised controlled trial looking to answer how the malaria infection affect the immune response for vaccines included
--

	in the expanded program for immunisation. The researchers teams will perform in vitro tests of immune response. it is a complex and well designed trial. However the limitations are no well discussed. Given that it is a communitary trial there are many unrecognized and uncontrolled factors (in example: dietary regimes changes, environmental changes) that can influence the results, this should be recognized. In addition the use of the combination dihydroartemisinin/piperaquine compulse to a vast number of exclusions. There are any estimation of the extent of these exclusions in the population to be studied? In any case, I consider that although its limitations, the project as planned will give important and crucial data, however it is necessary that researchers be aware of the limitations of the information to be obtained.
--	---

REVIEWER	Monica Cartelle Gestal Louisiana State University. Health Science Center, Shreveport. Shreveport Louisiana USA
REVIEW RETURNED	30-Jul-2020

GENERAL COMMENTS	This is a very well written and detailed protocol. My congratulations for such a beautiful job and the careful consideration of all the variables included int his study. I have only a couple of minor comments.  1. It was not clear to me what is the statistical analysis that will be performed once the samples are collected and the results are available, would it be a multivariate analysis? 2. When the samples are process, are commercially available kits going to be used or is instead a home made protocol? It will be very good to have more information detailing the assay to be run, as the results can vary between different kits or protocols. Overall it is a great protocol very well written and detailed.
--

REVIEWER	Joerg J Moehrle Medicines for Malaria Venture CH1225 Geneva Switzerland
REVIEW RETURNED	21-Sep-2020

GENERAL COMMENTS	The effect of intermittent preventive treatment for malaria with dihydroartemisinin-piperaquine on immune responses to vaccines among rural Ugandan adolescents: randomised controlled trial protocol B for the 'POPulation differences in VACCine responses' (POPVAC) programme The proposed study addresses an Interesting research question to understand better how parasitic disease can affect response to vaccines. The question whether intermittent or periodic antimalarial treatment with DHA/PQP affects the response to vaccinations could results in two outcomes: it appears the authors assume that antiparasitic treatment will increase the response to vaccination, it should also be discussed and analysed whether treatment with DHA/PQP could have a negative effect on immune responses. This aspect is not sufficiently discussed and could be addressed in the discussion where only potential cardiac risks associated with DHA/PQP treatment are mentione (line 323)
--

	The authors mention that DHA/PQP is currently not standard of care in Uganda, in this context It is even more important to state that DHA/PQP is currently not approved or recommended for intermittent preventive treatment and/or SMC, it should be clearly stated in the that this an experimental intervention : Line 76 “Parasites have long been proposed as modulators of vaccine responses” this claim should be more substantiated by providing references other than just the spz vaccine trial results. Immunomodulation through malaria has been investigated and more background information could be provided (e.g Kumar et al “Immunomodulation in Malaria DOI: https://doi.org/10.1007/978-1-4614-8757-9_136-1 Line 80: I suggest to reword this paragraph, stating first that there are three different studies in the framework of POPVAC a brief summary of the other protocols should be presented, followed by the explanation of protocol B otherwise the concept is difficult to understand. Line 104, I suggest that a single term for monthly DHA/PQP should be defined and used throughout the document, e.g. monthly preventive treatment, intermittent preventive treatment, currently several terms are used DHA/PQP has a significant food effect and the risk of causing cardiac AEs when high exposures are achieved, these exposures could be reached when taking a therapeutic dose of DHA/PQP with food. I could not find a reference that the dosing of DHA/PQP should be done under fasting conditions as per labelling (https://www.ema.europa.eu/en/documents/product-information/eurartesim-epar-product-information_en.pdf, page 3) Please clarify Table 1 and the administration of HPV vaccine: the table could be understood as if the first vaccination and 2nd boost is for all participants, whilst the first boost is only administered to girls only 195-198: I wonder why the response to BCG immunisation is assessed after 8 weeks, to the other vaccines after 4 weeks, is this an operational issue or based on the previously observed response to the different types of vaccines, should be explained in the protocol. Please provide reference that ig measurments for BCG are indicated at 8 weeks post immune whilst 4 weeks are sufficient for YF-17d etc (or was this 4 wk timeframe chosen for operational reasons)
--	--

VERSION 1 – AUTHOR RESPONSE

COMMENTS FROM REVIEWER 1

Comment 1

In the introduction, the choice of malaria as an infection that could affect vaccine responses is not well justified.

Response

The text has been expanded and references added to the introduction section to justify our investigation of malaria as an infection that could affect vaccine responses.

Changes in the manuscript: Line 93-101

Comment 2

What is the prevalence of malaria in Uganda in general, and in the study area in particular? It could be mentioned in the introduction

Response

The prevalence of malaria parasites in Uganda is quite variable, but parts of Uganda still have entomological inoculation rates among the highest in the world. Data have been added to the introduction section (line 84-87). Also, previous studies have reported malaria prevalence of > 50% among 5 to 15 year olds in the study area (Rehman et al. *Malar J* 2019;18(1):318. doi: 10.1186/s12936-019-2954-0 and Staedke et al. *The Lancet Global Health* 2018;6(6): e668-e79. doi: 10.1016/S2214-109X(18)30126-8). Line 263-264 has been edited to reflect this information.

Changes in the manuscript: Line 84-87, Line 263-264

Comment 3

Line 82: put "WHO"

Response

This has been done

Changes in the manuscript: Line 89

Comment 4

Line 319: It mentioned: "DP has not been adopted as standard of care either in Uganda or elsewhere." Why then choose this drug for treatment? If for some participants, you observe a failure in the treatment of malaria with this drug, would this not affect your expected results?

Response

We agree with the reviewer that DP has not been adopted as a standard of care either in Uganda or elsewhere; however, studies (including a Ugandan RCT) have shown that intermittent preventive treatment with DP is effective in preventing malaria (Rehman et al. *Malar J* 2019;18(1):318. doi: 10.1186/s12936-019-2954-0 and Staedke et al. *The Lancet Global Health* 2018;6(6): e668-e79. doi: 10.1016/S2214-109X(18)30126-8); therefore, we expect a benefit for malaria prevention with DP in this trial. These references have been cited in the manuscript (line 90-92). For such a study it is necessary to choose a treatment that is effective, but different, from the standard treatment regimens.

Changes in the manuscript: n/a

Comment 5

Line 310: Discussion. "Parasites have long been proposed as modulators of vaccine responses". Please add reference

Response

References have been added.

Changes in the manuscript: Line 332

COMMENTS FROM REVIEWER 2

Comment 1

This is randomised controlled trial looking to answer how the malaria infection affect the immune response for vaccines included in the expanded program for immunisation. The researchers teams will perform in vitro tests of immune response. it is a complex and well designed trial. However the limitations are no well discussed. Given that it is a communitary trial there are many unrecognized and uncontrolled factors (in example: dietary regimes changes, environmental changes) that can influence the results, this should be recognized. In addition the use of the combination dihydroartemisinin/piperaquine compulse to a vast number of exclusions. There are any estimation of the extent of these exclusions in the population to be studied? In any case, I consider that although its limitations, the project as planned will give important and crucial data, however it is necessary that researchers be aware of the limitations of the information to be obtained.

Response

We acknowledge and agree with the reviewer that environmental changes may mean varying malaria prevalence in the study area since malaria changes with season. We have discussed this limitation and how to mitigate it in the discussion section (line 358-361). Also, the primary analysis is planned to be restricted to those who have malaria infection at baseline. This will ensure we are investigating the effect of parasite removal. Since this is a randomised trial, we expect that several factors such as diet and environmental and seasonal changes will be similarly distributed between the trial arms. Furthermore, information on diet will be collected and may be used to check differences if any. Lastly, we do not expect a large number of individuals to be excluded as a result of using dihydroartemisinin/piperaquine for the study. In fact, in a Ugandan RCT investigating intermittent preventive malaria treatment with DP, a small proportion (0.34%) was excluded during screening due to reasons related to use of DP (Staedke et al. The Lancet Global Health 2018;6(6): e668-e79. doi: 10.1016/S2214-109X(18)30126-8). A statement about this has been added to the discussion section (line 355-356).

Changes in the manuscript: Line 355-356

COMMENTS FROM REVIEWER 3

This is a very well written and detailed protocol. My congratulations for such a beautiful job and the careful consideration of all the variables included in this study.

I have only a couple of minor comments.

Comment 1

It was not clear to me what is the statistical analysis that will be performed once the samples are collected and the results are available, would it be a multivariate analysis?

Response

We have edited the text in the manuscript to describe that unpaired t-tests will be used to assess differences in vaccine responses between the trial arms and that in the event of any imbalance in key

characteristics between trial arms, multivariable regression models will be used to adjust for these potential confounders. Also, a detailed analysis plan will be developed and made available on the online trial registration site before the study commences.

Changes in the manuscript: Line 321-325

Comment 2

When the samples are processed, are commercially available kits going to be used or is instead a home made protocol? It will be very good to have more information detailing the assay to be run, as the results can vary between different kits or protocols.

Response

We plan to use commercial kits for some of our primary outcome measurements (e.g. the BCG-specific IFN-gamma), and carefully optimised in-house assays utilising standard control sera in other instances, collaborating with other international laboratories for quality control. For example, HPV assays will be conducted in collaboration with the HPV Serology laboratory at the Frederick National Laboratory for Cancer Research, NIH, who will provide virus-like particles (VLPs) for HPV 16 and 18, provide standard sera and conduct QC for our assays. The Salmonella typhi LPS-specific IgG assays will be conducted in collaboration with Oxford University, utilising similar reagents and sharing QC material. The YF-17D plaque-reduction neutralisation test will be conducted by the arbovirology and influenza laboratory at the Uganda Virus Research Institute, a national reference laboratory for many emerging and re-emerging viral infections.

Changes in the manuscript: n/a

COMMENTS FROM REVIEWER 4

Comment 1

The proposed study addresses an interesting research question to understand better how parasitic disease can affect response to vaccines. The question whether intermittent or periodic antimalarial treatment with DHA/PQP affects the response to vaccinations could result in two outcomes: it appears the authors assume that antiparasitic treatment will increase the response to vaccination, it should also be discussed and analysed whether treatment with DHA/PQP could have a negative effect on immune responses. This aspect is not sufficiently discussed and could be addressed in the discussion where only potential cardiac risks associated with DHA/PQP treatment are mentioned (line 323)

Response

A statement has been added to the discussion section that although we hypothesise that treatment of malaria will improve vaccine responses, it is possible that it will negatively impact immune responses, or that effects will differ between types of vaccines.

Changes in the manuscript: Line 342-343, 351

Comment 2

The authors mention that DHA/PQP is currently not standard of care in Uganda, in this context it is even more important to state that DHA/PQP is currently not approved or recommended for intermittent preventive treatment and/or SMC, it should be clearly stated that this is an experimental intervention

Response

We thank the reviewer for this important comment. We acknowledge that DHA/PQP is currently not standard of care in Uganda or elsewhere and also not yet recommended for IPT of malaria; however, WHO guidelines (Geneva: World Health Organization. Guidelines for treatment of Malaria 2015; <https://www.who.int/malaria/publications/atoz/9789241549127/en/>) recommend it for treatment of malaria in pregnant women, infants, and young children. Furthermore, a randomized trial among Ugandan school children (Rehman et al. Malar J 2019;18(1):318. doi: 10.1186/s12936-019-2954-0) showed that IPT with DP reduces malaria infection. Also, a systematic review on Safety, tolerability, and efficacy of repeated doses of dihydroartemisinin-piperazine for prevention and treatment of malaria (Gutman et al., The Lancet Infectious diseases 2017;17(2):184-93. doi: 10.1016/s1473-3099(16)30378-4) concluded that monthly repeated doses are safe and efficacious. These references are cited in the manuscript line 90-92. We have now clearly stated in the manuscript that DP is an experimental but safe intervention.

Changes in the manuscript: Line 287-289

Comment 3

Line 76 "Parasites have long been proposed as modulators of vaccine responses" this claim should be more substantiated by providing references other than just the spz vaccine trial results. Immunomodulation through malaria has been investigated and more background information could be provided (e.g. Kumar et al "Immunomodulation in Malaria DOI: https://clicktime.symantec.com/32T3q8vkLpZQAoqCeKNBi4z6H2?u=https%3A%2F%2Fdoi.org%2F10.1007%2F978-1-4614-8757-9_136-1)

Response

We have added more background information in the introduction section showing that malaria can affect responses to several vaccines. Also, in the manuscript we mention immunological pathways through which malaria can impact vaccine responses (line 105-107)

Changes in the manuscript: Line 93-101

Comment 4

Line 80: I suggest to reword this paragraph, stating first that there are three different studies in the framework of POPVAC a brief summary of the other protocols should be presented, followed by the explanation of protocol B otherwise the concept is difficult to understand.

Response

A statement about the other POPVAC trials has been added.

Changes in the manuscript: Line 82-83

Comment 5

Line 104, I suggest that a single term for monthly DHA/PQP should be defined and used throughout the document, e.g. monthly preventive treatment, intermittent preventive treatment, currently several terms are used

Response

We thank the reviewer for this suggestion. The term 'Intermittent preventive treatment' has been adopted and used throughout the manuscript

Changes in the manuscript: Line 27,32,109,119, 287

Comment 6

DHA/PQP has a significant food effect and the risk of causing cardiac AEs when high exposures are achieved, these exposures could be reached when taking a therapeutic dose of DHA/PQP with food. I could not find a reference that the dosing of DHA/PQP should be done under fasting conditions as per labelling

(https://clicktime.symantec.com/3yAxkpaT7U2cWNcBBduMTM6H2?u=https%3A%2F%2Fwww.ema.europa.eu%2Fen%2Fdocuments%2Fproduct-information%2Feurartesim-epar-product-information_en.pdf, page 3)

Response

We have added a statement that high exposures to DP may be reached especially when taken with food.

Changes in the manuscript: Line 349-350

Comment 7

Please clarify Table 1 and the administration of HPV vaccine: the table could be understood as if the first vaccination and 2nd boost is for all participants, whilst the first boost is only administered to girls only

Response

It is true that the first HPV vaccination and 2nd boost will be given to all participants whereas week 8 HPV will only be given to girls ≥ 14 years to comply with the National EPI programme which recommends three doses of HPV for older girls. A footnote has been added to Table 1 to make this clear.

Changes in the manuscript: Footnote in Table 1

Comment 8

195-198: I wonder why the response to BCG immunisation is assessed after 8 weeks, to the other vaccines after 4 weeks, is this an operational issue or based on the previously observed response to the different types of vaccines, should be explained in the protocol.

Please provide reference that Ig measurements for BCG are indicated at 8 weeks post immune whilst 4 weeks are sufficient for YF-17D etc (or was this 4 wk timeframe chosen for operational reasons)

Response

References have been added to the manuscript to support sampling at 8 weeks post BCG and 4 weeks post YF-17D, Ty21a, HPV and Td as proposed for the primary endpoints, targeting the establishment of memory responses and approximate peak of antibody responses (Valentini et al., International Journal of Infectious Diseases 56 (2017) 140–154; Santos et al. <http://dx.doi.org/10.1590/S0074-02762005000300021>; Forrest Bruce, The Lancet, doi: 10.1016/s0140-6736(88)90284-x; Soares et al., The Journal of infectious diseases 2013;207(7):1084-94. doi: 10.1093/infdis/jis941; Lubyayi et al., Frontiers in Immunology 2020;11(929) doi: 10.3389/fimmu.2020.00929).

Changes in the manuscript: Line 213

VERSION 2 – REVIEW

REVIEWER	Odilon Paterne NOUATIN Centre de recherche médicales de Lambaréné (CERMEL / Gabon) Institut de Recherche clinique du Bénin 5IRCB / Bénin)
REVIEW RETURNED	30-Oct-2020

GENERAL COMMENTS	The manuscript is very well written, well structured, and the corrections are well made.
--

REVIEWER	Monica Cartelle Gestal LSU Health Science Center, Shreveport LA, USA
REVIEW RETURNED	06-Nov-2020

GENERAL COMMENTS	The manuscript has substantially improve, but a revision of the English is required as it still contain minor spelling errors.
--